# Metabolome Characteristics of Liver Autophagy Deficiency under Starvation Conditions in Infancy

**DOI:** 10.3390/nu13093026

**Published:** 2021-08-29

**Authors:** Kazuhito Sekiguchi, Hiroaki Miyahara, Masanori Inoue, Kyoko Kiyota, Kumiko Sakai, Toshikatsu Hanada, Kenji Ihara

**Affiliations:** 1Department of Pediatrics, Oita University Faculty of Medicine, Yufu, Oita 879-5593, Japan; sekiguch@oita-u.ac.jp (K.S.); m-inoue@oita-u.ac.jp (M.I.); k-sakai@oita-u.ac.jp (K.K.); 2Department of Neuropathology, Institute for Medical Science of Aging, Aichi Medical University, Aichi 480-1195, Japan; hmiya@oita-u.ac.jp; 3Institute for Research Promotion, Oita University Faculty of Medicine, Yufu, Oita 879-5593, Japan; kumi@oita-u.ac.jp; 4Department of Cell Biology, Oita University Faculty of Medicine, Yufu, Oita 879-5593, Japan; thanada@oita-u.ac.jp

**Keywords:** infant, autophagy, liver, starvation, metabolome

## Abstract

The liver function is essential for metabolism, detoxification, and bile synthesis, even in the neonatal period. Autophagy plays significance roles in THE adult liver, whereas the role of liver autophagy in the early neonatal period largely remains unclear. To clarify the importance of liver autophagy in the neonatal starvation period, we generated liver-specific autophagy-deficient (*Atg5^flox/flox^*; Albumin-Cre) mice and investigated under starvation conditions comparing with control (*Atg5^flox/+^*; Albumin-Cre) mice, focusing on serum metabolites and liver histopathology. As a result, autophagy in the liver was found to unessential for the survival under postnatal starvation. A metabolomics analysis of serum metabolites by gas chromatography-tandem mass spectrometry showed a significant difference between the groups, especially after 12-h starvation, suggesting the synergistical adaption of metabolic pathways, such as the “malate-aspartate shuttle”, “aspartate metabolism”, “urea cycle”, and “glycine and serine metabolism”. Liver-specific autophagy-deficiency under postnatal starvation conditions can cause a characteristic metabolic alteration suggesting a change of the mitochondrial function. Neonates seemed to maintain ketone production under starvation conditions, even in the autophagy-deficient liver, through a change in the mitochondrial function, which may be an adaptive mechanism for avoiding fatal starvation.

## 1. Introduction

The liver is the largest solid organ in the human body and plays significant roles in various biogenicities. The major functions of the liver include bile production, bilirubin metabolism, production of anti-coagulant factors and albumin, amino acid and fat metabolism, vitamin and mineral storage, massive blood filtering, and immunological reactions [1,2].

To sustain the liver function, autophagy plays crucial roles in these activities. Autophagy helps maintain the energetic balance of the liver through the turnover of different energy stores and the elimination of dysfunctional mitochondria. Autophagy also eliminates degenerated proteins and organelles that would otherwise become toxic for hepatocytes as a part of its function in controlling the cellular quality. Liver autophagy is activated as a defence against hepatic damage resulting from oxidative stress, organelle stress, or certain nutrients or xenobiotics [3,4,5,6,7].

Three different types of autophagy co-exist in the liver: macroautophagy, microautophagy, and chaperone-mediated autophagy [8,9]. The constitutive macroautophagy steadily contributes to the maintenance of hepatocyte homeostasis and protein quality control; proteins, organelles, and cytosolic materials are sequestered in a double-membraned vesicle to form an autophagosome that is subsequently degraded by lysosomal hydrolases, whereas inducible macroautophagy is activated during the first 4–6 h of starvation [10]. The increase in protein degradation in cytosol refills the pool of intracellular amino acids and thus preserves protein synthesis under starvation conditions. The amino acids generated by lysosomal breakdown can also enter the Krebs cycle and contribute to ATP production or be utilized for gluconeogenesis [11,12]. Adult mice with liver-specific deletion of autophagy demonstrated a failure to adjust to nutrient deprivation with support from starvation-induced proteolysis and lipolysis [13,14,15].

This autophagy-mediated adaptation to starvation has been suggested to be essential in neonates during the critical period from the interruption of the transplacental nutrient supply to breastfeeding from the mother. Autophagy in systemic tissues in neonatal mice were activated during the postnatal fasting period, and mice lacking the genes essential to autophagy died in the first days of life as a consequence of their inability to sustain a sufficient concentration of amino acids and synthesize essential proteins [16,17]. A recent study showed that the neuron-specific transgenic expression of *Autoghagy related 5* (*Atg5*) mice (*Atg5*^–^^/–^; *NSE*-*Atg5*) was able to prevent death in *Atg5*-null neonatal mice by enabling sucking behaviour. This fact indicated that stable feeding can maintain postnatal energy supply in the absence of liver autophagy. In contrast, the amino acid levels were reduced both in *Atg5*^–^^/–^ and *Atg5*^–^^/–^; *NSE*-*Atg5* neonates, suggesting that the restoration of *Atg5* in the brain did not improve the nutritional status [18]. We therefore believe that liver autophagy contributes to the synthesis of essential proteins, metabolizing bilirubin, detoxification of metabolites or drugs, and particularly the maintenance of nutritional homeostasis especially during starvation period just after birth.

To clarify the importance of liver autophagy in the neonatal nutritional crisis period, we generated hepatocyte-specific deficient of *Atg5* and intensively analysed newborns under starvation conditions, mainly focusing on serum metabolites using gas chromatography-tandem mass spectrometry (GC-MS/MS) and the structure of mitochondria using an electron microscope.

## 2. Materials and Methods

### 2.1. Animals

All animal handling and experiments were conducted according to the guidelines of the Animal Ethics Committee at Oita University and following the ARRIVE guidelines. All experimental protocols were approved by the Gene Recombination Safety Committee and the Animal Ethics Committee at Oita University. Liver-specific autophagy-deficient mouse models were generated using the Cre-loxP system. We used *Atg5*-floxed (*Atg5^flox/flox^*) mice [19] (RBRC02975; Riken BioResource Research Center, Ibaraki, Japan), homozygous transgenic Albumin-Cre mice (003574; Jackson Laboratory, Bar Harbor, ME, USA). To generate liver-specific autophagy-deficient mice (KO mice: *Atg5^flox/flox^*; Albumin-Cre), we first crossbred *Atg5^flox/flox^* females with homozygous Albumin-Cre males. The obtained *Atg5^flox/+^*; heterozygous Albumin-Cre females were then backcrossed to *Atg5^flox/flox^* males. Pregnant dams were euthanized, and mice including the KO mice and Control mice (*Atg5*^flox/+^; Albumin-Cre) were delivered at embryonic day (E) 18.5 by Caesarian section to prevent any breastfeeding. Euthanasia for pregnant mice by cervical dislocation was performed according to the *AVMA Guidelines for Euthanasia of Animals* (2020 edition). We used the following primers to detect wild-type Atg5 and Atg5flox alleles: A (exon3-1), 5′-GAATATGAAGGCACACCCCTGAAATG-3′; B (short2), 5′-GTACTGCATAATGGTTTAACT CTTGC-3′; C (check2), 5′-ACAACGTCGAGCACAGCTGCGCAAGG-3′ [19]. We used the following primers to detect transgenic Albumin-Cre alleles: A (Wild type Forward), 5′-TGCAAACATCACATGCACAC-3′; B (Common), 5′-TTGGCCCCTTACCATAACTG-3′; C (Mutant Forward), 5′-GAAGCA GAAGCTTAGGAAGATGG-3′.

### 2.2. Samples Collection

The birth weight of newborns with successful resuscitation was measured within 30 min after birth, and then animals were randomly assigned to 5 groups, with each group monitored in a thermostat chamber without feeding for 1, 3, 6, 9, and 12 h before being sacrificed. Blood samples were collected by decapitation. Euthanasia for newborns by decapitation was performed according to the *AVMA Guidelines for Euthanasia of Animals* (2020 edition). After storage for 1–2 h at 4 °C, the blood sample were centrifuged twice at 5000 rpm for 15 min and then once at 5000 rpm for 5 min to separate serum. The serum samples were stored at −70 °C. The tail and right upper and lower extremities were dissected and stored at −70 °C. The intraperitoneal organs were extracted by laparotomy at 4 °C. Liver tissues were used for protein extraction and immunohistochemical and electron microscopic analyses.

### 2.3. Protein Extraction and Western Blot Analyses

Separated livers were immediately homogenized in a 10-fold volume of radioimmunoprecipitation (RIPA) lysis buffer containing the protease inhibitor and phosphate inhibitors (SC-24948; Santa Cruz, Dallas, TX, USA). Samples were separated using sodium dodecyl sulfate-polyacrylamide gel electrophoresis (SDS-PAGE) on 10% Tris-HCl gels and then transferred to a polyvinylidene difluoride (PVDF) membrane. The expressions of NCoR1, PPARα and PINK1 were measured using the Automated Western Blots system (ProteinSimple Wes; ProteinSimple. San Jose, CA, USA). A protein analysis was performed following the protocol of the Wes ProteinSimple Kit and the approach described in previous report [20]. All reagents were prepared and used according to the manufacturer’s recommendations. The intensities of acquired chemiluminescence signals were quantified using the AlphaView and Compass software programs (ProteinSimple). Antibodies against ATG5 (NB110-53818; NOVUS Biologicals, Littleton, CO, USA; 1:1000), p62 (GP62-C; PROGEN. Heidelberg, Germany; 1:1,000), LC3 (PM036; Medical & Biological Laboratories, Nagoya, Japan; 1:500), NCoR1 (5948; Cell Signalling, Danvers, MA, USA; 1:10,000), PPARα (ab24509; abcam, Cambridge, UK; 1:50,000), PINK1 (6946; Cell Signalling; 1:10,000) and β-actin (A3845; Sigma-Aldrich, St. Louis, MO, USA; 1:50,000) were used as primary antibodies. The secondary antibodies used for Western blotting were anti-guinea pig conjugated to peroxidase (106-035-003, Jackson ImmunoResearch, West Grove, PA, USA; 1:10,000) and anti-rabbit conjugated to peroxidase (111-035-003, Jackson; 1:10,000). Blots were visualized by chemiluminescence.

### 2.4. Histopathological and Immunohistochemical Procedures

The specimens were fixed with 4% paraformaldehyde phosphate-buffered saline (PBS; 161-20141; Wako, Osaka, Japan) immediately after sacrifice and embedded in paraffin. Histological examinations were performed using 4.5-µm-thick sections stained with hematoxylin and eosin (HE). The paraffin-embedded sections were also visualized using N-Histofine simple stain MAX-PO (R) (724142; Nichirei Bioscience, Tokyo, Japan) and diaminobenzidine as the chromogen. As the primary antibodies, we used rabbit polyclonal antibodies against ATG5 (NB110-53818; NOVUS; 1:1000, pretreated by heating), LC3B (ab51520; abcam; 1:10,000, pre-treated by heating), and guinea pig polyclonal antibody against p62 (C-terminal) (GP62-C; PROGEN; 1:500, pre-treated by heating).

### 2.5. Metabolite Extraction and Derivatization

Metabolites were extracted using a modification of the Bligh-Dyer method [21]. A 40-μL aliquot of cold PBS was added to each tube of 10 μL of serum. When the total serum volume was <10 μL, PBS was added to compensate for the deficiency. A 5-μL aliquot of internal standard solution (2-isopropylmalic acid, 1 mg/mL) was added to the mixture. Methanol (500 μL) and 250 μL of chloroform were added to each sample at 4 °C followed by mixing with a vortex mixer for 30 s. Next, 250 μL of Milli Q water (Merck, Darmstadt, Germany) was added to each sample followed by mixing with a vortex mixer for another 30 s. The mixtures were then centrifuged at 14,000 rpm for 6 min at room temperature. A 500-μL aliquot of the supernatant was transferred to new tubes and dried in an Eppendorf concentrator at room temperature for 3 h. The fully dried samples were then combined with 80 μL of methoxyamine hydrochloride in pyridine (20 mg/mL), mixed with a vortex mixer for 30 s, and sonicated for 30 s until the solids were fully dispersed. The samples were shaken at 30 °C for 90 min at 200 rpm (BioShaker; Taitec Co., Saitama, Japan). After shaking, 80 μL of N-methyl-N-trimethylsilyltrifluoroacetamide was added to the samples, followed by further shaking at 37 °C for 30 min at 200 rpm using the BioShaker. Finally, the derivatized samples were centrifuged at 14,000 rpm for 5 min at room temperature, and 100 μL of the supernatant was transferred to GC-MS vials using micro tubes.

### 2.6. Metabolomic Analyses

The analysis of metabolites was performed by GC-MS/MS. The GC-MS/MS analysis was performed on a GCMS-TQ8040 system (Shimadzu Corporation, Kyoto, Japan) equipped with a DB-5 capillary column (30 m × 0.25 mm inner diameter, film thickness 1 μm; Agilent, Santa Clara, CA, USA). Each 1-μm aliquot of the derivatized sample solution was automatically injected in splitless mode into the gas-liquid chromatography column using an auto-injector (AOC-20i; Shimadzu Corporation). During the GCMS-TQ8040 analysis, the injector temperature was kept at 280 °C, and helium was used as a carrier gas at a constant flow rate of 39.0 cm/s. The GC column temperature was programmed to remain at 100 °C for 4 min and then to rise to 320 °C at a rate of 10 °C/min, holding at 320 °C for a further 11 min. The ionization voltage was 70 eV. Argon was used for collision-induced dissociation. Metabolite detection was performed using the Smart Metabolites Database Ver. 3 software program (Shimadzu Corporation) using the method described in a previous study with some modifications [22]. The 2-isopropylmalic acid contained in the extraction solution was also used to evaluate the stability of our GC-MS/MS analysis system. Peak identification was performed automatically and then confirmed manually based on the specific precursor and product ions as well as the retention time using the method described in a previous study [23,24]. Selected data are represented as box-and-whisker plots and were compared using the Mann-Whitney U test with the SPSS Statistics 23.0 software program (SPSS Statistics for Windows, IBM, New York, NY, USA). *p* < 0.05 was considered statistically significant.

### 2.7. Multivariate Statistical Analyses

The integral metabolomics datasets were imported into the SIMCA version 13.0.3.0 software program (Umetrics, Umea, Sweden) for multivariate statistical analyses. OPLS-DA with Pareto scaling was used to visualize the differences between the metabolomics data sets and extract the significant metabolites. The S-plot visualizes both the covariance and correlation between the metabolites and modelled class designation. The compounds with an absolute value of *p* (corr) > 0.7 in the S-plot were selected as significant metabolites. In order to determine the pathways altered between metabolomics data sets, MetPA and MSEA with the significant metabolites were performed using the MetaboAnalyst 4.0 software program (https://www.metaboanalyst.ca/, accessed on 14 August 2020) for the pathway analysis [25,26]. The function of the metabolic pathways judged to be significant were confirmed using the SMPDB (http://www.smpdb.ca, accessed on 19 March 2021) [27].

### 2.8. Electron Microscopic Analyses

The morphological changes in the ultrafine structure of mitochondria in hepatocytes were observed under an electron microscope after 1 and 12 h of starvation. For the electron microscopic analysis, the liver tissues were fixed with 2% glutaraldehyde in phosphate buffer (0.2 M, pH 7.4) overnight at 4 °C and post-fixed with 1% OsO4 (124505; Merck) in distilled water for 2 h at room temperature. After dehydration through graded ethanol washes, the samples were embedded in Epon 812. Ultrathin sections were cut with an ultramicrotome (Reichet Ultracut S; Leica, Wetzlar, Germany), stained with aqueous uranyl acetate and lead citrate, and observed in transmission electron microscope (TEM) (H-7650; Hitachi, Tokyo, Japan) at an acceleration voltage of 80 kV. The mitochondrial structural abnormalities of mitochondria were determined with reference to the abnormality type charts previously reported [28]. Image analysis was performed with open-source image-analysis programs WCIF ImageJ (developed by W. Rasband). The length (major axis), width (minor axis), mitochondrial aspect ratio (ratio between major and minor axes of an ellipse equivalent to the mitochondrion as an index for mitochondrial morphology, minimum value is 1.0) and area of each mitochondrion was measured as mitochondrial morphologic indicators [29,30]. Selected data are represented as the mean ± standard error of the mean and were compared using Student’s *t*-test. Statistical analyses were performed using the SPSS Statistics software program (version 23.0, IBM). *p* values of <0.05 were considered statistically significant.

## 3. Results

### 3.1. Birth Weight and Mortality Did Not Change with Liver Autophagy

To examine the effects of liver autophagy on birth weight and the starvation survival, liver-specific autophagy-deficient mice (*Atg5^flox/flox^*; Albumin-Cre: KO group) and control mice (*Atg5^flox/+^*; Albumin-Cre: Control group) were obtained by Caesarean section at E18.5 from pregnant *Atg5^flox/+^*; heterozygous Albumin-Cre females mating with *Atg5^flox/flox^* males. Genotyping was performed by polymerase chain reaction (PCR) using cryopreserved tail tissue obtained from euthanized neonates. Genotyping of 196 newborns revealed that 50 (25.5%) were *Atg5^flox/flox^*; Albumin-Cre (KO group), 45 (23.0%) were *Atg5^flox/+^*; Albumin-Cre (Control group), 53 (27.0%) were *Atg5^flox/flox^*, and 48 (24.5%) were *Atg5^flox/+^*. The birth weight of KO mice (1.184 ± 0.02413 g [mean ± SEM], *n* = 50) was not significantly different from that of Control mice (1.210 ± 0.02578 g [mean ± SEM], *n* = 45) (Figure 1A). The physical activity and external appearance of the KO newborns were also not markedly different from those of the Controls (Figure 1B). Most of the newborns in both groups survived, and we started observation under starvation conditions up to 22 h after birth (Figure 1C).

### 3.2. Interference with the Liver Autophagic Activity under Starvation Conditions Was Noted in the KO Group

To assess the autophagic activity in the liver under starvation conditions, we analysed the expression of autophagy markers, including cytosolic form of microtubule associated protein 1 light chain 3 (LC3-I), phosphatidylethanolamine conjugate form of LC3 (LC3-II), and p62, by Western blotting. The ATG5 expression of the liver had nearly been completely depleted in KO mice (Figure 2A). LC3-I was highly expressed, and the LC3-II/LC3-I ratio was significantly lower in the KO mice during starvation than in the Control mice (Figure 2B). The p62 expression was high in KO mice (Figure 2C). An immunohistochemical evaluation also showed that the ATG5 expression and autophagosome structures of microtubule-associated protein 1A/1B-light chain 3B (LC3B) were decreased, whereas the dot-like-structures of p62 were significantly increased in the liver of KO mice during starvation, indicating the deficient status of liver autophagy (Figure 2D–F).

### 3.3. NCoR1 and PPARα Levels in Liver

We measured the nuclear receptor co-receptor 1 (NCoR1) and peroxisome proliferator-activated receptor α (PPARα) expression using an automated Western blot system. The expression of NCoR1 tended to be higher in liver-specific autophagic KO mice than in Control mice after 12 h of starvation. The PPARα was expressed in all of the analysed mice, and there was no significant difference in the expression of PPARα between the KO and Control mice (Appendix A). 

### 3.4. Liver Autophagy Affected Neonatal Serum Metabolome

More than three newborns were sacrificed at each point of 1, 3, 6, 9, 12 h of starvation, and each serum sample was analysed for metabolites by GC-MS/MS. As a result, 320 types of metabolites were discovered, and 252 of those were detected in all samples.

#### 3.4.1. Carbohydrates

Glucose, galactose and fructose levels showed no remarkable decreases during starvation. There were no significant differences between the groups. The glycerol level in the KO group was significantly lower, whereas the lactose level in the KO group was higher than that in the Control group at 12 h of starvation (Figure 3A).

#### 3.4.2. Ketone Bodies

Regarding the three types of ketone bodies we measured, there was no increase in the production of ketones in newborns of the Control group during 12 h of starvation and no significant differences were observed between the groups (Figure 3B). This suggests that the production of ketone bodies was maintained in the neonatal liver during starvation, even without autophagy in liver.

#### 3.4.3. Components of the Tricarboxylic Acid Cycle

Succinic acid or fumaric acid, the components of the tricarboxylic acid (TCA) cycle, decreased with starvation in the Control group, whereas these were significantly higher in the KO group at 12 h of starvation (Figure 3C).

#### 3.4.4. Free Fatty Acids

The serum levels of most free fatty acids (FFAs) were relatively high in the KO group during the starvation, regardless of the length of the carbon chain or saturation status (Figure 4A). Significant differences were observed after 12 h of starvation in four of 14 FFAs (Figure 4B). FFAs are derived from the degradation of triglycerides, which are released from adipose tissue by hormone-sensitive lipases and taken up into the liver for glucose and lipid metabolism. Saturated odd-chain fatty acids, such as docosahexaenoic acid (C17:0), are not naturally generated in animals or plants; thus, those in the serum were likely transplacentally transferred from the mother. The noted differences between the groups suggest that the uptake and/or metabolism of FFAs in the liver seemed to be changed in the autophagy-deficient liver under postnatal starvation.

#### 3.4.5. Amino Acids

The comprehensive analysis of amino acid levels demonstrated that most were high in the KO group during starvation, including phenylalanine, cysteine, or asparagine (Figure 4C,D), probably due to the wide-ranging alteration of amino acid metabolism in autophagy-deficient liver to adapt to starvation conditions. 

The amounts of aromatic amino acids (phenylalanine, tryptophan) and sulphur-containing amino acids (cysteine and methionine) tended to differ more markedly between the groups in comparison to aliphatic amino acids (alanine, isoleucine, leucine, proline, and valine) or glycine.

We further observed the state of metabolism under a longer starvation period up to 22 h. We detected a marked decrease in glucose and galactose levels in survivors of both groups (Appendix A). No significant differences in the levels of ketone bodies, metabolites of the TCA cycle, FFAs or most amino acids were observed between the groups at 22 h of starvation. (Appendix A). These results suggested that metabolic adaptation to starvation was only effective for a short period (about 12 h) of starvation after birth.

### 3.5. Acceleration of Autophagic Activity of Non-Liver Tissues Was Not Observed in KO Group by Starvation

The autophagic activity of tissues other than the liver was evaluated. Focusing on skeletal muscle tissues (Appendix A), we found no significant difference in the LC3-II/LC3-I between the groups, whereas the expression of p62 was higher in the KO mice than in the Control mice. As a result, these data did not support the acceleration of autophagy in skeletal muscle of the liver-specific autophagy-deficient mice to compensate for the deficient state of liver metabolism.

### 3.6. Lack of Liver Autophagy Caused Alteration in Metabolism as Revealed by a Multivariate Analysis of Metabolites

The orthogonal partial least squares discriminant analysis (OPLS-DA) model was applied for the multivariate analysis of metabolites. The Scores scatter plots at each starvation time are shown in Figure 5A (12 h) and Appendix A (1, 3, 6 and 9 h, respectively).

The Scores scatter plot of the OPLS-DA model showed that the metabolic profile of the KO group was segregated from the Control group with acceptable R2X (0.694–0.989) and positive Q2 value (0.0238–0.433) at each starvation duration (Appendix A). S-plots enabled the visualization and filtering of significantly altered metabolites with liver-specific autophagy deficiency (Figure 5B, Appendix A). The numbers of metabolites with a potential impact as biomarkers selected from the S-plot with |p (corrected)| >0.7 at 1, 3, 6, 9, and 12 h starvation were 2, 2, 10, 0, and 52 metabolites, respectively (Appendix A).

We then searched for candidates of metabolic pathways contributing to the liver-specific autophagy in starvation using two analysis algorithms (metabolic pathway analysis [MetPA] and metabolite set enrichment analysis [MESA]) listed in Metaboanalyst, a free tool for conducting pathway analyses. As a result, we identified 3 and 6 pathways after 6 and 12 h of starvation, respectively. At 6 h of starvation, “Fructose and mannose degradation” was extracted as the liver autophagy-related pathway with both analytical algorithms. “Lactose degradation” and “galactose metabolism” were determined to be metabolic pathways showing significant differences based on the presence or absence of liver-specific autophagy. These pathways would synergistically operate under an increased glucose demand. At 12 h of starvation, “ammonia recycling” and “malate-aspartate shuttle” were extracted as significant metabolic pathways with the MetPA algorithm (Figure 5C), and “aspartate metabolism”, “urea cycle” and “glycine and serine metabolism” were extracted by the MESA algorithm (Figure 5D, Table 1).

### 3.7. Structural Difference in the Mitochondria of the Liver under Postnatal Starvation Conditions between Autophagy-Deficient Mice and Control Mice

To investigate the effect of postnatal starvation in autophagy-deficient hepatocytes, we studied the morphology of the ultrafine structure of mitochondria in hepatocytes under an electron microscope. Number of observed mice/cells/mitochondria from Control and KO group at 1 h starvation were 3/6/147 and 3/6/172, respectively. Number of observed mice/cells/mitochondria from the Control and KO group at 12 h starvation were 3/6/157 and 3/6/136, respectively. As a result, morphological changes of mitochondria, such as altered cristae and aberrant inclusions, were observed in KO mice after 12 h of starvation (Figure 6). The length, width, aspect ratio, and mitochondrial area of KO mice were not significant different from Control mice at 1 h of starvation (length, 0.9341 ± 0.02557 µm [mean ± SEM] vs. 0.9576 ± 0.02722 µm; width, 0.6127 ± 0.01332 µm vs. 0.6550 ± 0.01744 µm; aspect ratio, 1.592 ± 0.04988 vs. 1.551 ± 0.05390; mitochondrial area, 0.4831 ± 0.01937 µm^2^ vs. 0.5282 ± 0.02542 µm^2^, respectively). The mitochondrial diameter (length, width) of KO mice at 12 h was longer than Control mice (length, 1.060 ± 0.04300 µm vs. 0.8982 ± 0.03297 µm, p=0.012; width, 0.6396 ± 0.02215 µm vs. 0.4863 ± 0.01205 µm, *p* < 0.001, respectively). The aspect ratio (length/width) of mitochondria from KO mice was lower than Control mice (1.806 ± 0.08498 vs. 1.937 ± 0.09287, *p* = 0.011). The mitochondrial area of KO mice was larger than Control mice (0.6037 ± 0.04626 µm^2^ vs. 0. 3835 ± 0.02121 µm^2^, *p* < 0.001) (Figure 6B–E). These data suggest that the altered pattern of the metabolome would be causally related to the morphological difference of the mitochondria during the neonatal period under starvation conditions between liver-specific autophagy-deficient mice and Control mice.

We measured the PTEN-induced putative kinase 1 (PINK1) expression in liver using an automated Western blot system. There was no apparent difference in PINK1 expression between KO and Control mice (Appendix A).

## 4. Discussion

Liver autophagy in newborns is involved in the metabolism of carbohydrates, amino acids, fatty acids, and ketone bodies under starvation conditions. First, we confirmed that autophagy in the liver was not essential for the survival of newborns under postnatal starvation conditions, being primarily compensated by the adaptive alternation of amino acid metabolism for glyconeogenesis. This result was consistent with the findings of previous papers, wherein newborn mice with the neuron-specific transgenic expression of ATG5 avoided lethality due to *Atg5* deficiency [31]. Since the stock of lipid, protein, and glycogen in neonates is markedly smaller than in adults [32,33], we speculated that starvation-induced macroautophagy in non-liver organs or tissues might play a compensatory role in maintaining energy production in neonates.

In the metabolome data, the serum concentration of alanine, which constitute a majority of the glycogenic amino acids in serum, gradually declined over 12 h of starvation in both KO and Control mice. Thus, the amino acids, probably produced by catabolism in skeletal muscles, seemed to be used as the major energy resource of alanine for conversion to glucose (“alanine shuttle”). According to the Small Molecule Pathway Database (SMPDB), the “malate-aspartate shuttle” system in mitochondria is essential for allowing electrons to move across the impermeable membrane between the cytosol and mitochondrial matrix. These electrons are created during glycolysis and are crucial for oxidative phosphorylation. The “ammonia recycling” pathway contributes to the reuse of ammonia for amino acid synthesis. Regarding “aspartate metabolism”, aspartate is also a metabolite in the urea cycle and is involved in gluconeogenesis. In addition, aspartate carries the reducing equivalents in the mitochondrial malate-aspartate shuttle, which utilizes the ready interconversion of aspartate and oxaloacetate. The “glycine and serine metabolism” pathway is involved in the synthesis and breakdown of small amino acids, including glycine, serine, and cysteine, and these compounds share common intermediates, functioning as a part of the “ammonia recycling” pathway. The “homocysteine degradation” pathway constitutes a part of the methionine metabolic pathway, and homocysteine in combination with serine yields L-cysteine, ammonia, and 2-oxobutanoate, which is a TCA cycle intermediate. The “urea cycle” is a major metabolic pathway involved in hepatic urea production and amino acids, including arginine, citrulline, and ornithine, in cytosol or mitochondria. The pathway software program did not identify “fatty acid metabolism/oxidation” in the MetPA algorithm, probably because the GC-MS/MS system in this study did not detect acyl-carnitines or acetyl-CoA-bound FFAs. All of the identified pathways play separate key roles in metabolism under starvation conditions. We therefore hypothesize that mitochondria may be deeply involved in each pathway, such as detoxification of ammonia in the urea cycle, oxidative deamination of glutamate, energy production by the citric acid cycle from pyruvate, which is generated via the amino acid catabolism of Ala, Gly, Ser, Cys, Thr, and Trp and electron transfer by the malate-aspartate shuttle.

A previous study of adult mice demonstrated the accumulation of NCoR1 and PPARα reduction in autophagy-deficient liver [14]. The changes of NCoR1 and PPARα after 12 h of starvation in autophagy-deficient newborns were milder than those in adults. In addition, the ketone production was not decreased, which was consistent with the previous studies, which showed that neonatal ketogenesis is activated independently of starvation [32,34].

The structural difference of liver mitochondria under postnatal starvation condition between KO mice and Control mice was observed, whereas mitophagy was not detected in our neonatal study. The previous study described that mitochondria actively elongated to avoid autophagic degradation and sustained cell viability under low nutrients [35,36], whereas mitophagy was typically observed more than 24 h of starvation [37]. Although we did not biochemically analyse the beta-oxidation in the mitochondria, we estimated that morphological differences in mitochondria under starvation condition represented the differences in the energy production of mitochondria between KO mice and Control mice. Taken together, the functional change of mitochondrial metabolism with normal ketone production may be an adaptive mechanism for avoiding fatal starvation after birth, regardless of the autophagy function in the neonatal liver.

We found elevated levels of mitochondrial metabolites in the TCA cycle, such as succinic acid or fumaric acid, in liver-autophagy-deficient neonates. These findings are similar to those obtained using human serum data under conditions of severe bacterial infection [38,39]. We propose that it may be possible to predict autophagic dysfunction mimicking bacterial infection based on the elevation of serum levels of mitochondrial metabolites.

In conclusion, liver-specific autophagy-deficiency under postnatal starvation conditions can cause a characteristic metabolic state under an altered mitochondrial function. Neonates can produce ketones, even under autophagy-deficient starvation conditions in the liver, which may be an adaptive mechanism for avoiding fatal nutritional crisis. Further research is needed to evaluate the clinical utility of the metabolites, such as serum succinic acid, as a biomarker for predicting autophagy dysfunction in the liver that mimics an infection state.

## Figures and Tables

**Figure 1 nutrients-13-03026-f001:**
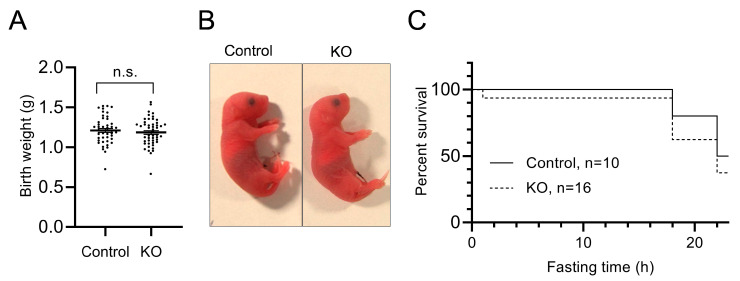
The phenotype at birth and early postnatal lethality with starvation of liver-specific Atg5-deficient and Control newborn mice. (**A**) The birth weight of KO (*Atg5*^flox/flox^; Albumin-Cre+) and Control (*Atg5^flox/+^*; Albumin-Cre+) mice. The birth weight (mean ± SEM) was not significantly different between the groups. The number of mice were 50 (KO) and 45 (Control). (**B**) Photograph of a representative KO mouse compared with a Control littermate. No abnormal appearance was noted in the KO or Control newborns. (**C**) Kaplan-Meier curves of KO and Control mice. There was no significant difference in the survival rates up to 22 h of starvation. KO: liver-specific *Atg5*-deficient; n.s.: not significant.

**Figure 2 nutrients-13-03026-f002:**
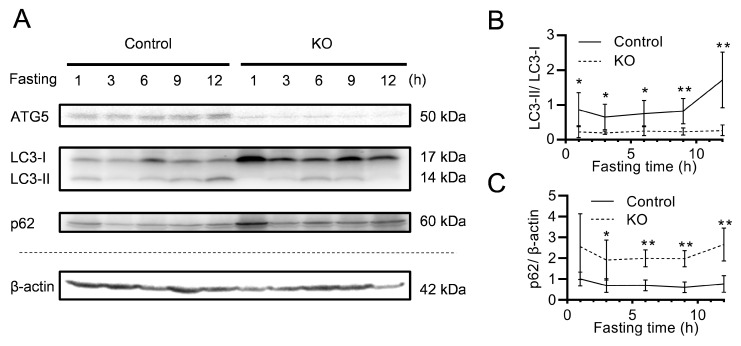
The Western blotting and immunohistochemical analyses of ATG5, LC3, and p62 in liver extract. (**A**) The protein expression of the ATG5, LC3 and p62 in liver extract after 1, 3, 6, 9, and 12 h starvation. β-actin was proceeded in parallel gel and used as an internal control. (**B**,**C**) The relative LC3-II/LC3-I ratio (**B**) and p62 level (**C**) are shown in graphs. The numbers of mice (starvation time) were as follows: 5 (1 h), 5 (3 h), 5 (6 h), 5 (9 h), and 6 (12 h) Control mice and 4 (1 h), 5 (3 h), 5 (6 h), 5 (9 h), and 5 (12 h) KO mice. Each error bar is expressed as the mean ± SEM. Student’s *t*-test; * *p* < 0.05, ** *p* < 0.01. (**D**) ATG5, LC3B and p62 immunostainings of liver samples of KO mice and Control mice after 12 h starvation. (**E**) Liver cells with intracytoplasmic dot-like structures labelled by anti-LC3B antibody. The LC3B expression was decreased in KO mice (**F**) Dots labelled by anti-p62 antibody. The expression of p62 were significantly increased during starvation in KO mice. Samples were immunostained and then counterstained with hematoxylin. Bar = 100 µm for ATG5, 10 µm for LC3B, and 20 µm for p62. KO: liver-specific *Atg5*-deficient; ATG5: autophagy related 5; LC3: microtubule-associated protein 1A/1B-light chain 3; LC3-I; cytosolic form of LC3; LC3-II: phosphatidylethanolamine conjugate form of LC3; LC3B: LC3 microtubule-associated protein 1A/1B-light chain 3B; HPF: high power fields.

**Figure 3 nutrients-13-03026-f003:**
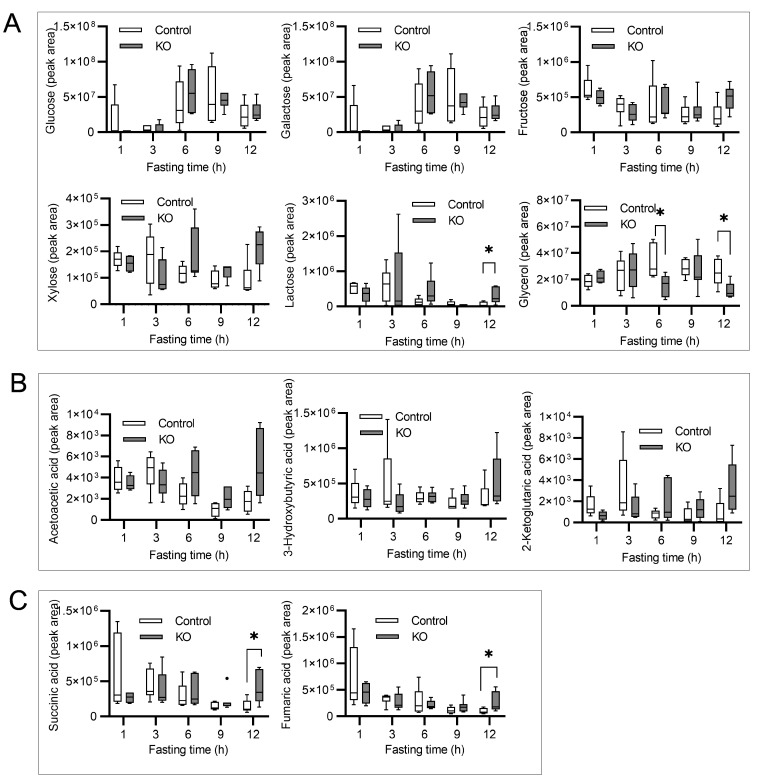
The gas chromatography-tandem mass spectrometry analysis of metabolites in the serum of starved neonatal mice. The numbers of mice (starvation time) were as follows: 5 (1 h), 6 (3 h), 5 (6 h), 6 (9 h), and 6 (12 h) Control mice and 4 (1 h), 5 (3 h), 7 (6 h), 7 (9 h), and 5 (12 h) KO mice. The metabolite concentrations were measured as the peak area of the calibration curve, and the concentration at each starvation time was compared between the groups. The levels of serum saccharides and glycerol (**A**), ketone bodies (**B**), and representative metabolites of the TCA cycle (**C**) during starvation. Boxes represent the interquartile range (25th to 75th percentiles), and lines within the boxes are the median; error bars represent the 25th percentile minus 1.5 times the interquartile range (IQR) and the 75th percentile plus 1.5 times the IQR. Mann-Whitney U test; * *p* < 0.05, KO: liver-specific *Atg5*-deficient.

**Figure 4 nutrients-13-03026-f004:**
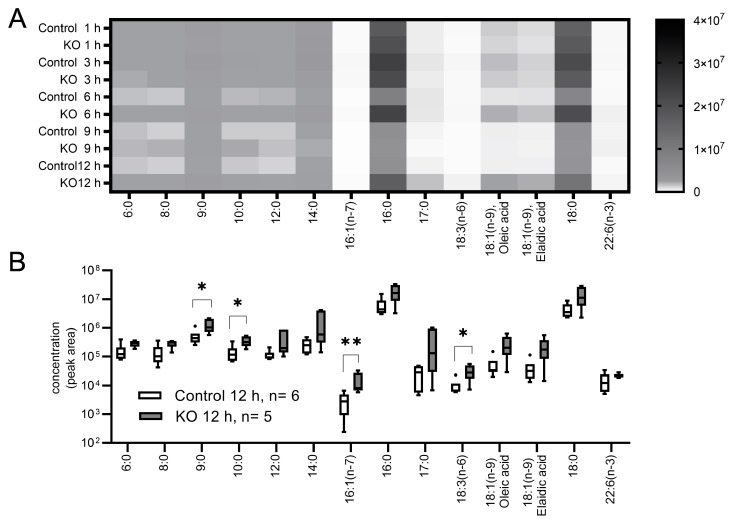
Serum FFA and amino acids concentrations in starved mice according to a gas chromatography-tandem mass spectrometry analysis. The numbers of mice (starvation time) were as follows: 5 (1 h), 6 (3 h), 5 (6 h), 6 (9 h), and 6 (12 h) Control mice and 4 (1 h), 5 (3 h), 7 (6 h), 7 (9 h), and 5 (12 h) KO mice. The metabolite concentrations were measured as the peak area of the calibration curve, and each metabolite after 12-h of starvation was compared between the groups. (**A**) Heat map of changes in the concentration of 14 FFAs. Higher density matrixes indicate a larger amount of FFAs. (**B**) FFA concentrations after 12-h of starvation. Boxes represent the interquartile range (25th to 75th percentiles), and lines within the boxes are the median; error bars represent the 25th percentile minus 1.5 times the interquartile range (IQR) and the 75th percentile plus 1.5 times IQR. (**C**) Heat map of changes in the concentration of 20 amino acids. Higher density matrixes indicate a larger amount of amino acids. (**D**) Amino acid concentrations after 12-h of starvation. Boxes represent the interquartile range (25th to 75th percentiles), and lines within the boxes are the median; error bars represent the 25th percentile minus 1.5 times the interquartile range (IQR) and the 75th percentile plus 1.5 times IQR. Mann-Whitney U test; * *p* < 0.05, ** *p* < 0.01. KO: liver-specific *Atg5*-deficient.

**Figure 5 nutrients-13-03026-f005:**
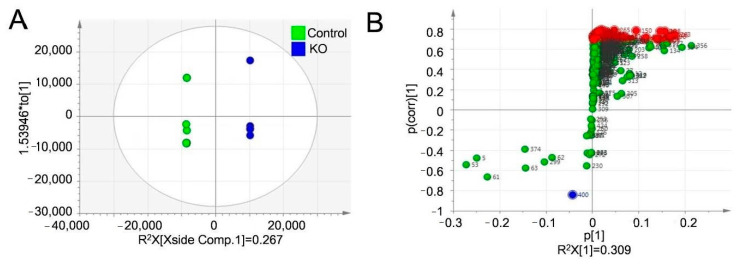
Multivariate statistical and pathway analyses of serum metabolites after 12 h of starvation. (**A**) The OPLS-DA score scatter plot generated from serum metabolite data of KO and Control mice. The *x*-axis shows the intra-group variance, and the *y*-axis shows the inter-group variance. Each point represents the results of an analysis of data from one mouse. A differential metabolic pattern was noted between KO (blue) and Control group (green) by the plots clearly separated on the score scatter plot. (**B**) S-plot of serum metabolites from KO and Control mice. Each dot indicates individual metabolite. The numbers beside the dots indicate the individual numbers of metabolites. The S-plot visualizes both the covariance (*y*-axis) and correlation (*y*-axis, *p*-value) between the metabolites and the modelled class designation. The 52 significantly altered metabolites with 0.7 < *p* (corr) <1.0 (red dots) or –1.0 < *p* (corr) < –0.7 (blue dot) were selected as potential biomarker metabolites (shown in Appendix A.). (**C**) A MetPA based on potential biomarker metabolites. The MetPA visualizes the pathway impact (*y*-axis), *p*-value (*y*-axis, colour of the circle), and number of hit metabolites (width of circle). (**D**) A MSEA based on potential biomarker metabolites. Color intensity (yellow to red) reflects increasing statistical significance MSEA. The lengths of the bars represent the fold enrichment. Six pathways were identified as significantly changed metabolic pathways affected by liver-specific autophagy-deficiency (Ammonia Recycling, Malate-Aspartate Shuttle, Homocysteine Degradation, Aspartate Metabolism, Urea Cycle and Glycine and Serine Metabolism). KO: liver-specific *Atg5*-deficient; OPLS-DA: orthogonal partial least squares discriminant analysis; MetPA: metabolic pathway analysis; MSEA: metabolite set enrichment analysis.

**Figure 6 nutrients-13-03026-f006:**
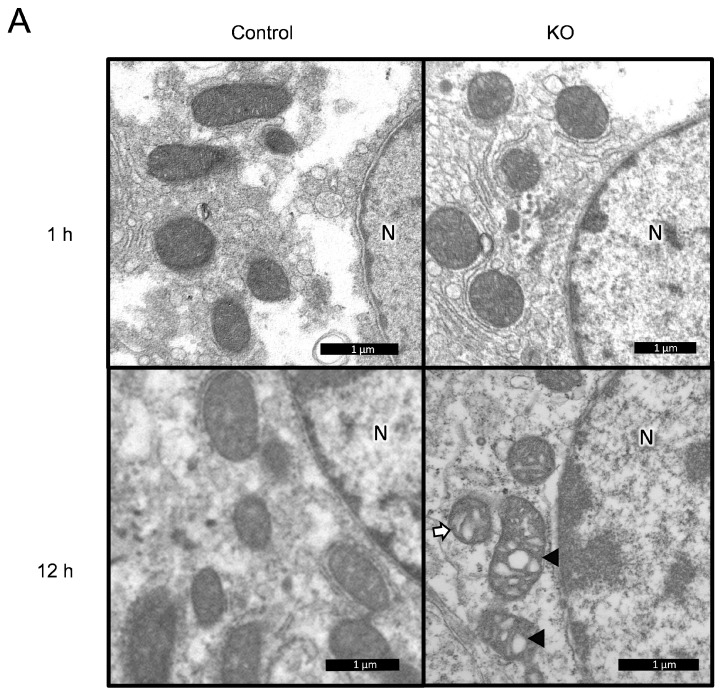
Ultrastructural demonstration of mitochondria in hepatocytes of Control and KO mice after 1 and 12 h of starvation. (**A**) Representative structural findings of mitochondria. After 1 h of starvation, the mitochondrial structures were almost normal in both KO and control mice. Damaged mitochondria, such as those with dysmorphic cristae and abnormal inclusion, were noticed in KO mice after 12 h of starvation. Bar = 1 µm. N: nucleus. (**B**–**E**) Quantification of mitochondrial length (**B**), width (**C**), aspect ratio (**D**) and area (**E**) in hepatic cells of Control and KO group at 1 h and 12 h starvation (mean ± SEM). Number of mice/cells/mitochondria of Control and KO group at 1 h starvation were 3/6/147 and 3/6/172, respectively. Number of mice/cells/mitochondria of Control and KO group at 12 h starvation were 3/6/157 and 3/6/136, respectively. Student T test; * *p* < 0.05, ** *p* < 0.01, *** *p* < 0.001. KO: liver-specific *Atg5*-deficient.

**Table 1 nutrients-13-03026-t001:** The corresponding pathways deprived by Pathway Analysis from serum metabolites of OPLS-DA analysis between KO and Control group at 6 h and 12 h starvation.

Method	Starvation Duration	Pathway Name	Total	Expected	Hits	*p*
MetPA	6 h					
		Fructose and Mannose Degradation	28	0.2775	2	0.029132
	12 h					
		Ammonia Recycling	25	1.0654	4	0.018828
		Homocysteine Degradation	7	0.29832	2	0.032521
		Malate-Aspartate Shuttle	7	0.29832	2	0.032521
MSEA						
	6 h					
		Lactose Degradation	9	0.0879	2	0.00298
		Galactose Metabolism	38	0.371	3	0.00473
		Fructose and Mannose Degradation	32	0.312	2	0.0364
	12 h					
		Malate-Aspartate Shuttle	10	0.42	3	0.00674
		Ammonia Recycling	32	1.34	5	0.00891
		Aspartate Metabolism	35	1.47	5	0.0131
		Urea Cycle	29	1.22	4	0.0298
		Glycine and Serine Metabolism	59	2.48	6	0.0324

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
