# Peer review of "Metabolome Characteristics of Liver Autophagy Deficiency under Starvation Conditions in Infancy"

_nutrients, 2021, doi:10.3390/nu13093026_

Round 1
Reviewer 1 Report
In this research, the authors clarify that neonates with liver-specific autophagy-deficiency under postnatal starvation conditions can produce ketones through altered mitochondrial function and may be an adaptive mechanism for avoiding fatal nutritional crisis. The authors focus on analyze and compare the expression of metabolites between liver-specific Atg5 knockout and wild-type mice. The authors clearly investigate the role of autophagy in neonates under starvation by liver-specific autophagy-deficient (Atg5flox/flox; Albumin-Cre) mice system, however, there are some concerns as following:
- The authors measured the nuclear receptor co-receptor 1 (NCoR1) and demonstrated that the expression of NCoR1 is higher in liver-specific autophagic KO mice than in Control mice after 12 h of starvation, suggesting that degradation of NCoR1 is impaired by autophagy deficiency in neonates (line 254). How about mRNA expression in KO and control mice? The authors need to provide the mRNA expression of NCoR1 because without the mRNA expression, it is hard to conclude that NCoR1 expression increased is through impaired autophagy degradation process.
- Under homeostatic conditions, the mitochondrial network is continuously shaped by fission and fusion. Dysfunctional mitochondria are degraded by mitophagy. Does mitophagy occur in the liver of control mice under starvation?
- In this research, the authors conclude that liver-specific autophagy-deficiency under postnatal starvation conditions can cause a characteristic metabolic state under an altered mitochondrial function. The length and width of mitochondria in the KO mice are longer than control. Mitochondria actively elongate to avoid autophagic degradation and sustain cell viability under low nutrients [Curr Biol. 2011 Jun 21;21(12):R478-80]. Please discuss how Atg5KO further increases the length and width of mitochondria in the authors’ mouse system.
- Color intensity (yellow to red) reflects increasing statistical significance in metabolite set enrichment analysis (MSEA). The author made the wrong description in figure legends (Fig 5D; line 397-398), please correct it.
- The resolution of all data is low (the reviewer is not sure whether because of converting PDF file), please improve it.
Reviewer 2 Report
The manuscript entitled Brain activity of thioctic Acid enantiomers in vitro and in vivo studies in an animal model of care is well done. My concerns mainly regard the methodological aspects.
The authors should significantly change the images in figure 2. The quality of them is so low! Besides, they should add the legend to figure the calibration bar measure. This information is put on the images, but unfortunately, they are not readable.
In section 3.7, the authors explain the structural changes in the mitochondria of the autophagy-deficient liver under postnatal starvation conditions. Unfortunately, they omit important information. In the structural study, the presence of imaging analysis and adding some photos of the electronic microscope (in this case) are essential. Therefore, they have to add this information otherwise their analysis may seem like a mathematical consideration only but not a morphological investigation.
Round 2
Reviewer 2 Report
The authors have addressed my concerns.